# TELL ME WHY!—EXPLANATIONS SUPPORT LEARNING OF RELATIONAL AND CAUSAL STRUCTURE

## ABSTRACT

Explanations play a considerable role in human learning, especially in areas that remain major challenges for AI—forming abstractions, and learning about the relational and causal structure of the world. Here, we explore whether machine learning models might likewise benefit from explanations. We outline a family of relational tasks that involve selecting an object that is the *odd one out* in a set (i.e., unique along one of many possible feature dimensions). Odd-one-out tasks require agents to reason over multi-dimensional relationships among a set of objects. We show that agents do not learn these tasks well from reward alone, but achieve $> 90\%$ performance when they are also trained to generate language explaining object properties or why a choice is correct or incorrect. In further experiments, we show how predicting explanations enables agents to generalize appropriately from ambiguous, causally-confounded training, and even to meta-learn to perform experimental interventions to identify causal structure. We show that explanations help overcome the tendency of agents to fixate on simple features, and explore which aspects of explanations make them most beneficial. Our results suggest that learning from explanations is a powerful principle that could offer a promising path towards training more robust and general machine learning systems.

Explanations—language that provides explicit information about the abstract, causal structure of the world—are central to human learning (Keil et al., 2000; Lombrozo, 2006). Explanations help solve the credit assignment problem, because they link a concrete situation to generalizable abstractions that can be used in the future (Lombrozo, 2006; Lombrozo and Carey, 2006). Thus explanations allow us to learn efficiently, from otherwise underspecified examples (Ahn et al., 1992). Human explanations selectively highlight generalizable causal factors and thereby improve our causal understanding (Lombrozo and Carey, 2006). Similarly, they help us to make comparisons and master relational and analogical reasoning (Gentner and Christie, 2008; Lupyan, 2008; Edwards et al., 2019). Even explaining to ourselves, without feedback, can improve our ability to generalize (Chi et al., 1994; Rittle-Johnson, 2006; Williams and Lombrozo, 2010), potentially because explanations abstract knowledge, thus making it easier to recall and generalize (cf. Dasgupta and Gershman, 2021).

These abilities—abstraction, causality, relations and analogies, and generalization—are often cited as areas where deep learning has not yet achieved human-level performance (e.g. Fodor and Pylyshyn, 1988; Lake et al., 2017; Pearl, 2018; Marcus, 2020; Ichien et al., 2021; Holyoak and Lu, 2021; Puebla and Bowers, 2021; Geirhos et al., 2020). Thus, there has been increasing interest in using explanations as learning signal (e.g. Ross et al., 2017; Mu et al., 2020; Camburu et al., 2018; Schramowski et al., 2020; see related work). That is, rather than seeking explanations only post-hoc, in an attempt to help humans understand the system (e.g. Chen et al., 2018; Topin and Veloso, 2019; Xie et al., 2020), these works use explanations to help the system understand the task (cf. Santoro et al., 2021). However, most research on learning from explanations has been within supervised learning. There have been a few explorations within reinforcement learning (RL; Guan et al., 2021; Tulli et al., 2020), but these have not focused on relational or causal reasoning. Yet explanations may be particularly useful in these contexts, to augment the sparse learning signals of rewards. Furthermore, explanations help humans to learn causal structure, and RL agents can learn causal structure (e.g. Dasgupta et al., 2019), while supervised learners cannot (Pearl, 2018). We therefore explore the benefits of language explanations for RL tasks involving relational and causal structure.

We define a language explanation to be a string that indicates relationships between a situation, the agent's behavior, and abstract task structure. Thus, explanations can highlight aspects of a

situation that are generalizable to other instances of that task, by elevating task-relevant features above idiosyncratic or task-irrelevant ones. We show that language explanations improve agent learning and generalization, and explore how different aspects of explanations affect their benefits.

We first outline a set of challenging relational tasks involving uniqueness—identifying the object from a set that is the *odd-one-out* along some dimension. These tasks are difficult to learn, because agents need to consider the relationships among all the present objects along various dimensions, rather than simply the properties of a single object or even pairs of objects. We show that these tasks are difficult for agents to learn from reward alone in multiple environments and learning paradigms

However, learning and generalization improve substantially when agents are also trained to generate language explanations—even without prior knowledge of language and without telling agents how to use the explanations. Explanations highlight abstract task structure, and thus outperform task-agnostic auxiliary losses (even ones that offer more supervision). Explanations allow agents to learn more effectively, without fixating on easy-but-inadequate "shortcut" features (Geirhos et al., 2020; Hermann and Lampinen, 2020). Explanations allow agents to disentangle confounded features and generalize appropriately out-of-distribution to deconfounded evaluation, and to meta-learn to perform interventions to identify causal structure. We then explore different how different aspects of explanations contribute to their benefits.

Our results suggest that generating explanations could be a powerful tool for learning challenging RL tasks. Language explanations may be simpler for humans to produce than other forms of supervision (e.g. Cabi et al., 2019; Guan et al., 2021), and could selectively identify the key generalizable features of a situation. Thus, training agents to generate explanations might provide a path towards improving learning and generalization. We highlight the following as our main contributions:

1. The family of cognitively-relevant *odd-one-out* tasks (§1) provide challenging measures of agents' ability to extract abstract relations, across various settings and paradigms.
2. Generating explanations (§2) substantially improves RL agent learning on these tasks (§3)—even without prior knowledge of language and without telling agents how to use explanations. Explanations selectively highlight the generalizable structure of a particular task, yielding better performance than less-selective unsupervised auxiliary losses.
3. Explanations can disentangle confounded features from ambiguous training (§3.2).
4. Explanations help agents learn to perform experiments to identify causal structure (§3.3).
5. Explanations help agents move beyond simple biases. Explanations that respond to agent behavior are best, and outputting explanations is better than receiving them as input (§3.4).

# 1 THE ODD-ONE-OUT TASKS

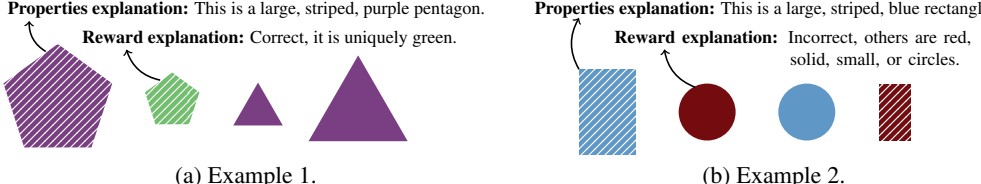

(a) Example 1.           (b) Example 2.

Figure 1: Conceptual illustrations of two possible odd-one-out tasks, and corresponding possible explanations. This figure depicts odd-one-out tasks with feature dimensions of color, texture, shape, and size, and the two types of explanations we consider. Property explanations identify relevant object features, while reward explanations specify which feature(s) make the choice correct or incorrect. (a) The second object is the odd one out, because it is a unique color. (b) The first object is the odd one out, because it is uniquely large. Explanations of incorrect choices identify all features.

We outline a challenging family of fundamentally-relational tasks: finding the odd one out in a set of objects, i.e. the one that is somehow unique (Fig. 1). Odd-one-out tasks have been used extensively in cognitive science (e.g. Stephens and Navarro, 2008; Crutch et al., 2009), and proposed in perceptual settings in robotics (Sinapov and Stoytchev, 2010). These tasks are challenging, because they involve both relational reasoning (same vs. different) and abstraction (identifying uniqueness requires reasoning over all objects, and all dimensions along which objects may be related). Furthermore, these tasks permit informative explanations, of relevant dimensions, properties, and relations.

Investigating these challenging and abstract—yet explainable—relational tasks is particularly interesting, because relational reasoning and abstraction are critical human abilities (Gentner, 2003; Penn et al., 2008), but the capacity of deep learning to acquire these skills is disputed (e.g. Santoro et al., 2017; 2018; Geiger et al., 2020; Ichien et al., 2021; Puebla and Bowers, 2021). However, explanations are important to human relational learning (Gentner and Christie, 2008; Lupyan, 2008; Edwards et al., 2019), suggesting that explanations might similarly help machines acquire these skills.

In Fig. 1 we conceptually illustrate some odd-one-out tasks. In Fig. 1a the second object is a uniquely green, while the rest are purple. We thus denote color as the *relevant* dimension in this episode. Along the other, irrelevant dimensions—shape, texture, and size—attributes appear in pairs. For example, there are two pentagons and two triangles. These pairs force the agent to consider *all* the objects. If the agent considered only the first three objects it would be unable to tell whether the first object was the odd one out (uniquely large), the second (uniquely green), or the third (uniquely a triangle or uniquely solid textured). Thus, the agent must consider all objects to identify the correct dimension and the correct unique feature. This makes the relational reasoning particularly challenging, since the agent must consider many possible relationships. The agent is rewarded for selecting the odd-one-out, either by picking it up, or naming it, depending on the task instantiation.

We emphasize that in principle these tasks can be learned from reward alone—language is not necessary for performing them, so we evaluate without language. Nevertheless, we find that in practice language explanations are critical for learning these tasks in our settings. We consider two types of explanations: reward explanations and property explanations (see Fig. 1 for examples). Reward explanations are produced after the agent chooses, and identify the feature(s) that make the choice correct or incorrect. Property explanations are produced before the agent chooses, and explain the identity of the object the agent is facing by specifying its task-relevant properties. Both types satisfy our above criterion for explanations: they link the situation and the agent's behavior to the task structure.

## 1.1 ENVIRONMENTS

Odd-one-out tasks can be instantiated in various settings, from games to language or images, and can incorporate various latent structures (e.g. meta-learning). We instantiate these tasks in 2D and 3D RL environments (Fig. 3a). In 2D, the agent has simple directional movement actions, while in 3D it can move, look around, and grasp nearby objects at which it is looking. In both environments we place an agent in a room containing four objects, which vary along feature dimensions of color, texture, position, and either shape (2D) or size (3D). In each episode, one object will be unique along one dimension. The 3D environment compounds the difficulty of the odd-one-out tasks, because the agent's limited view often forces it to compare objects in memory. See Appx. C.2 for full details.

## 2 METHOD: GENERATING EXPLANATIONS AS AUXILIARY TRAINING

We focus on language explanations provided during training. We synthetically generate the explanations online, conditional on agent behavior. However, explanations could be produced by humans, e.g. as annotations of past trajectories (cf. Ross et al., 2017). We train the agent to generate explanations as an auxiliary signal to shape its representations (Fig. 2), as opposed to providing explanations as direct inputs (which is less effective; Appx. A.3); our approach thus does not require explanations at test time. Note that we do not directly supervise behaviour through explanations, nor tell the agent how to use them. The agent simply predicts explanations as an auxiliary output.

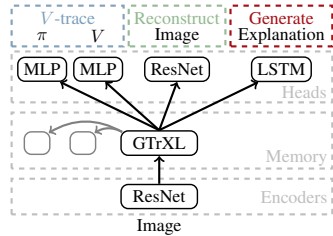

Figure 2: RL agent with auxiliary prediction of explanations.

We train agents using the IMPALA (Espeholt et al., 2018) framework. Our agent (Fig. 2) consists of a visual encoder, a memory, and output heads. The encoder is a CNN or ResNet (task-dependent). The agent remembers using a 4-layer Gated TransformerXL Memory (Parisotto et al., 2020), with the visual encoder output and previous reward as inputs. The output of the memory is input to the heads. The policy and value heads are MLPs, trained with $V$-trace. Another head reconstructs the input images, to learn better representations (though this is not necessary; Appx. A.7). Finally, the explanation head is a single-layer LSTM, which generates language explanations. We train the agent to predict these explanations using a summed softmax cross-entropy loss. See Appx. C.1 for details.

# 3 EXPERIMENTS

## 3.1 PERCEPTUAL ODD-ONE-OUT TASKS IN 2D AND 3D RL ENVIRONMENTS

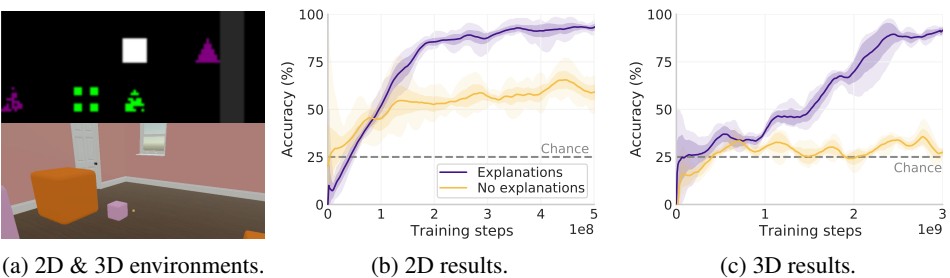

(a) 2D & 3D environments.  (b) 2D results.  (c) 3D results.

Figure 3: Explanations help agents learn the perceptual odd-one-out tasks in both RL environments. (a) Our environments in 2D (top) and 3D (bottom). In 2D, the agent is the white square, while in 3D it has a first-person view. The objects appear in varying positions, colors, textures, and shapes (2D) or sizes (3D). (b) 2D results. Agents trained with explanations achieve high performance; agents trained without do not. (c) 3D results. Only agents trained with explanations learn the tasks. (Training steps denotes actor/environment steps, number of parameter updates is $\sim 10^4 \times$ smaller. 5 seeds per condition in 2D, 3 per in 3D, lines=means, dark region=$\pm$SD, light region=range.)

**Explanations help agents to learn perceptual odd-one-out tasks.** We first evaluate the benefit of explanations for learning the odd-one-out tasks, by comparing agents trained without explanations to agents trained with property and reward explanations. In both 2D and 3D environments, agents trained with explanations learn to solve the tasks over 90% of the time (Figs. 3b-c). Agents trained without explanations perform worse; in the easier 2D environment they exhibit partial learning (see 3.4), while in the more challenging 3D environment they barely perform above chance. All agents in 2D were trained with an auxiliary unsupervised reconstruction loss. However, agents trained without reconstruction but with explanations perform well (Appx. A.7), while agents trained with reconstruction but without explanations do not. Because explanations highlight abstract task structure, they outperform task-agnostic unsupervised objectives, even ones that provide strictly more supervision.

## 3.2 EXPLANATIONS CAN DECONFOUND

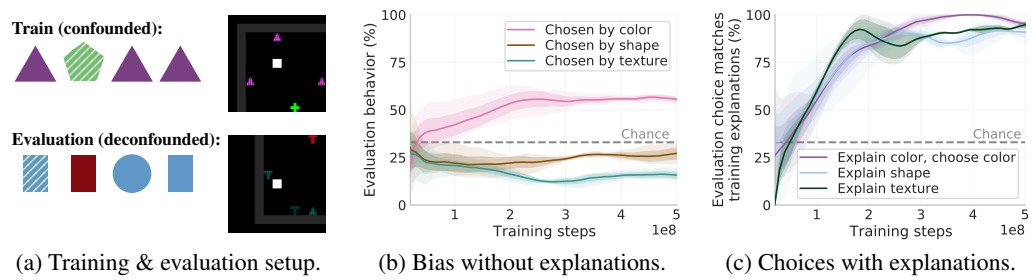

(a) Training & evaluation setup.  (b) Bias without explanations.  (c) Choices with explanations.

Figure 4: Explanations can deconfound perfectly correlated features. (a) Schematic depictions and environment screenshots from train and test. The agent is trained in confounded settings, where the target object is unique in color, shape, and texture. The agent is tested in deconfounded settings, where one object is unique along each dimension (and an additional distractor object has no unique attributes). (b) When trained without explanations, the agent is biased towards using color (the simplest feature) in evaluation. (c) However, if the agent is trained with explanations that target any particular feature, the agent tends to use that feature to choose in the deconfounded evaluation. (3 seeds per condition, chance is random choice among valid objects.)

For humans, explanations help identify *which specific aspects* of a situation are generalizable (Lombrozo and Carey, 2006). Could explanations also help RL agents to disentangle confounded features, and generalize to out-of-distribution tests? We explore this with a different training and testing setup (Fig. 4a). In training, one object is the odd-one-out along *three* feature dimensions (color, shape, and

texture). Thus, any or all of these features could be used to solve the task—the dimensions are perfectly confounded. In test, however, the features are deconfounded: there is a different odd-one-out along each dimension. We explore the effect of explanations that consistently refer to a single feature dimension (without mentioning others) on the agent's behavior in deconfounded evaluation. We train agents in four conditions: no explanations, color-only explanations, shape-only explanations, or texture-only explanations. Although color, shape, and texture are confounded within each episode, their values appear in different combinations across episodes—e.g. all triangles are purple in Fig. 4a, but in another episode they may all be red. Thus, single-dimension explanations can potentially draw the agent's attention to a particular dimension, and thereby disentangle these features, even though the explanations do not alter the relationship between these dimensions and the reward signal.

We found that agents trained without explanations were biased towards using color (the simplest feature) in the deconfounded evaluation (Fig. 4b). However, the agents trained with explanations generalized in accordance with the dimension that they were trained to explain $> 85\%$ of the time (Fig. 4c), even though there were no direct cues linking the reward to that dimension over the others. In this setting, shaping an agent's internal representations through explanations draws its attention to the desired dimension, and allows $> 85\%$ out-of-distribution generalization along that dimension.

## 3.3 EXPLANATIONS HELP AGENTS META-LEARN TO EXPERIMENT TO IDENTIFY CAUSAL STRUCTURE

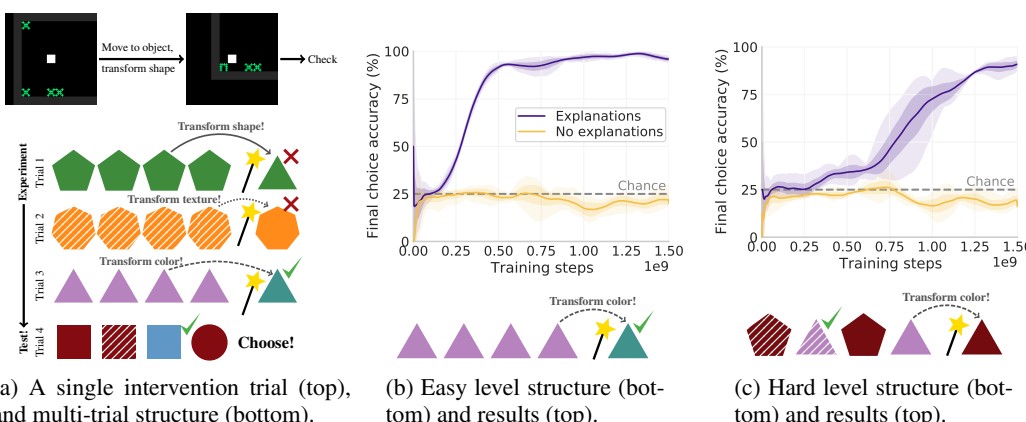

(a) A single intervention trial (top), and multi-trial structure (bottom).

(b) Easy level structure (bottom) and results (top).

(c) Hard level structure (bottom) and results (top).

Figure 5: Explanations allow agents to meta-learn to perform experiments. (a) Each episode consists of 3 trials where the agent gets to experiment with a magic wand in order to discover which feature dimension is relevant, followed by a final trial where it must choose the unique object along that dimension. In this case the relevant dimension is color. In the first trials the agent transforms the shape and texture of the objects, but is not rewarded for picking them up. In the third trial, it transforms the color and is rewarded for picking the object up. The agent can then infer that it should choose the different-colored object in the final trial. (b) In some episodes, the experiments are easy, because all the object attributes are the same, and the agent only needs to transform an object and select that object. Agents trained with explanations learn these tasks, while agents trained without explanations do not. (c) In other episodes, the experiments are harder, because the object attributes are all paired—the agent must transform one object, and then pick up *another* which has been made unique. With explanations, agents learn these difficult levels as well. (4 seeds per condition.)

Explanations help humans to understand causal structure (Lombrozo, 2006; Lombrozo and Carey, 2006). The ability of deep learning to learn causal structure is sometimes questioned (e.g. Pearl, 2019), but while theoretical limitations hold for passive learners, RL agents can intervene and can therefore learn causal structure. Indeed, agents can meta-learn causal reasoning in simple settings (Dasgupta et al., 2019). We therefore investigate whether explanations could help agents meta-learn to identify causal structure in more challenging odd-one-out tasks in richer environments.

We consider a meta-learning setting where agents complete episodes with four trials each. On each trial, the agents perform the odd-one-out task. There is only one causally important dimension per episode—reward is determined by uniqueness on only one of the feature dimensions (e.g. color).

The "correct" dimension is not directly observable, so the agents must learn to *perform experiments* on the first three trials to identify the causally relevant dimension, in order to select the correct object on a fourth test trial (Fig. 5a). The agent receives 1 reward for completing an early trial correctly, but 10 reward for completing the final trial correctly. Thus, the agent is incentivized to experiment and discover the correct dimension in the early trials, in order to gain a large reward in the final trial.

In the first three trials of each episode, we give the agent a magic wand that can perform one causal intervention per trial: changing an object's color, shape, or texture. The agent is forced to use the wand to create an odd-one-out, because each trial's initial configuration lacks objects with unique features—along each dimension the features are either all the same, or appear in pairs. When the features are all the same, the experiments are relatively easy (Fig. 5b): the agent must simply transform an object and then select the same object. When the features are all paired, however, the experiments are harder (Fig. 5c): the agent must transform one object, which will change to match other objects *and then it must select a different object* that was *formerly* paired with this one, but is now unique. The final trial is always a deconfounded test, where a different object is unique along each dimension, and with no access to the magic wand. On all trials, we reward the agent only if it selects an object which is unique along the "correct" dimension. Thus, the agent cannot consistently choose correctly unless it has already experimented with the magic wand to infer the correct dimension.

We again compare agents that receive property and reward explanations to agents without, but in this case the explanations are augmented to identify the correct dimension (e.g. "incorrect, the dimension is shape, and other objects are squares"). We find that agents trained without explanations cannot learn these tasks, while agents trained with explanations achieve high success at both easy (Fig. 5b) and harder levels (Fig. 5c). We find that in these complex, intervention-focused tasks, behaviourally-relevant explanations are necessary (Fig. 8b-c). Furthermore, reward explanations are necessary for any learning to occur, but having property explanations in addition is helpful (Fig. 6c).

## 3.4 Exploring the benefits of explanation in more detail

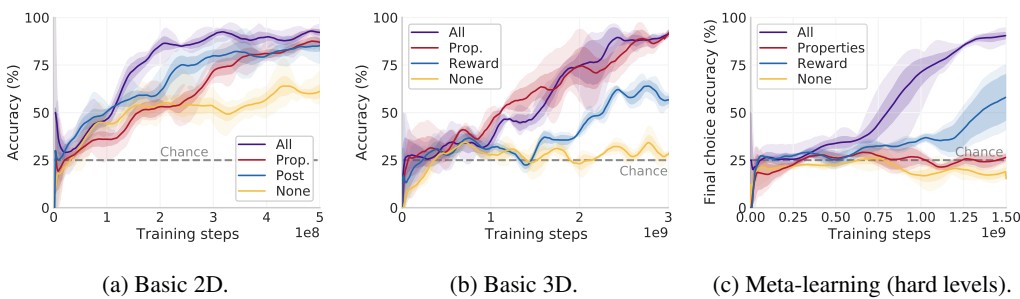

|  (a) Basic 2D. | (b) Basic 3D. | (c) Meta-learning (hard levels). |

Figure 6: Different explanation types offer complementary, separable benefits. We compare agents trained with all explanations or none (as above) to those trained with only property explanations (red), or only reward explanations (blue). (a) In the basic 2D tasks, either kind of explanations is sufficient for learning, but having both types together is substantially faster. (b) In the 3D tasks, property explanations result in comparable learning, while reward explanations are not as effective, but still better than none. This is likely due to the memory challenge of these tasks—property explanations can help the agent discover what to encode to make its choice, while reward explanations cannot. (c) In the meta-learning to experiment setting, by contrast, only reward explanations (or both together) result in any learning on the hard levels. (2 seeds per sub-type condition.)

In order to better understand the benefits of explanations, we explored our results further in a variety of analyses, ablations, and control and experiments. We outline our findings here; see Appx. A for full results. We found that explanations as input are not helpful (Appx. A.3), and can interfere with the benefits of explanations as targets. We also found that a curriculum of tasks that teaches object properties is not as effective as explanations (Appx. A.6), and that explanations are more beneficial in more complex tasks (Appx. A.5). We highlight three particularly interesting findings here.

**Explanations help agents overcome biases toward easy features (Appx. A.1):** In 2D, agents without explanations fixate on positions and colors, and learn to solve the task only when those

dimensions happen to be relevant. Shape and texture are generally not learned at all. This explains the moderate performance without explanations. With explanations, by contrast, agents learn to solve the tasks with any feature. Similarly, in the confounded features setting color is preferred without explanations, but again with explanations agents can learn to use other features. Hermann and Lampinen (2020) show similar feature-difficulty rankings for CNNs, and that CNNs lazily prefer easier features. Similarly, Geirhos et al. (2020) discuss "shortcut features" that networks prefer, despite the fact that those features do not correctly solve the task. Thus, explanations may help an agent to overcome biases towards easy-but-inaccurate solutions, to more completely master the task.

**Both explanation types provide complementary benefits; their relative value depends on the environment (Fig. 6):** In the above experiments we provided agents with both property and reward explanations. Here, we compare to agents trained to generate only a single type of explanations. We found that having both types of explanations is generally better than a single type, but the relative benefits of different types depend on the setting. In the 2D environment (Fig. 6a) either type of explanations alone results in learning, but both types together result in substantially faster learning. In the 3D setting (Fig. 6b), we find that property explanations are relatively more beneficial; perhaps because predicting explanations on encountering the objects helps the agent overcome the memory challenges in the 3D environment by helping it to encode the relevant features in an easily decodable way. By contrast, for the meta-learning tasks (Fig. 6c), we find that the reward explanations are necessary for any learning. The likely reason for this is clear when considering the episode structure— the relevance of a transformation to the final reward is much more directly conveyed by the reward explanations than the property ones. However, both types of explanations together are required for complete learning within the training budget we considered. In summary, the relative benefits of the explanation types depend on the demands of the environment, but generally having both types is best.

**Behaviourally- and contextually-relevant explanations are most effective (Appx. A.2):** Human explanations are *pragmatic* communication—they depend on context, knowledge, and behavior (Van Fraassen, 1988). We therefore compared to control explanations that referred to objects in the room, but independent of behaviour (on 10% of steps we randomly chose an explanation that could occur in the current room, regardless of agent actions), and irrelevant explanations (randomly sampled from those possible in *any* room). We found that behavior-relevant explanations were much more beneficial than behavior-irrelevant ones, and completely irrelevant explanations had no benefit. Explanations should respond to the agents actions rather than passively conveying information.

## 4 RELATED WORK

Explanations play many roles in human learning, such as enabling efficient learning, even from a single example (Ahn et al., 1992). Explanations play several roles: highlighting both causal factors, and relationships between a present situation and broader principles (Lombrozo, 2006). They therefore depend strongly on prior knowledge, and the relationship between explainer, the recipient, and the situation to be explained (Van Fraassen, 1988; Cassens et al., 2021). As Wood et al. (1976) say: "one must recognize the relation between means and ends in order to benefit from 'knowledge of results.'" Explanations link a specific situation to more general principles that can be used in the future.

**Relations:** Relational and analogical reasoning are often considered crucial to human intelligence (Gentner, 2003), and possibly absent in other animals (Penn et al., 2008). The relations *same* and *different* are central to many of these accounts, but their origins are heavily disputed (Penn et al., 2008; Katz and Wright, 2021, e.g.). But language and culture likely play a critical role in learning these concepts and skills (Gentner and Christie, 2008; Lupyan, 2008)—"relational concepts are not simply given in the natural world: they are culturally and linguistically shaped" (Gentner, 2003). Thus, explanations may be particularly key to these abilities. This may explain disagreements over neural networks' capacity for relational reasoning (Geiger et al., 2020; Puebla and Bowers, 2021; Ichien et al., 2021), at least without relational inductive biases (Santoro et al., 2017).

**Causality:** Humans focus on causal structure, even as children (Gopnik et al., 1999; Gopnik and Sobel, 2000), and our causal understanding is closely linked to explanations (Lombrozo and Vasilyeva, 2017). Human explanations are not just causal, but emphasize important causal factors that are useful for future prediction and intervention (Lombrozo and Carey, 2006). Furthermore, Lombrozo and Carey (2006) emphasize that children accept various explanations, while adults selectively endorse causally generalizable ones, suggesting that this focus may be at least partly learned.

**Self-explanation:** Asking humans to produce explanations for themselves, without providing feedback, can improve generalization (e.g. Chi et al., 1994; Rittle-Johnson, 2006; Williams and Lombrozo, 2010). Furthermore, Nam and McClelland (2021) find that the ability to produce explanations is strongly related to the ability to learn a generalizable problem-solving strategy involving relational reasoning. Furthermore, education—especially in formal mathematics—is related to developing these abilities. Are the ability to generate explanations and generalize (meta-)learned together?

## 4.1 RELATED WORK IN AI

We are certainly not the first to observe that human studies like the above suggest that explanations might help in AI—there has been a variety of prior work on explanation in AI, which we review here. We also relate to the broader set of approaches for auxiliary supervision that help agents (or models) to learn more effective representations for a task. Explanations are a particularly targeted form of auxiliary supervision that focuses on the causally-relevant, generalizable elements of a situation.

**Language as representation, or to shape representations?** Andreas et al. (2018) used language as a latent bottleneck representation in meta-learning, and found benefits. However, Mu et al. (2020) showed that it was better to *not* bottleneck through language, but merely use descriptions to shape latent representations in supervised classification tasks. This is closely related to our approach, but we evaluate varied explanations in more challenging tasks, across a broader range of settings.

**Natural Language Processing:** Explanations fit naturally into NLP tasks, and Hase and Bansal (2021) highlight the many ways that explanations could enter in NLP tasks, e.g. as targets, inputs, or as priors. Surprisingly, they find no improvement from using explanations as targets in their survey. However, they show some positive effects of explanation retrieval during both training and test, including improved performance on relational tasks and better disentangling of causal factors.

**Feature explanations as learning tools:** Some prior work has refined models using input attention or gradients as targets for explanatory feedback (e.g. from humans). Ross et al. (2017) show that penalizing gradients to irrelevant features can improve generalization on a variety of image and language tasks. Lertvittayakumjorn and Toni (2021) survey works on tuning NLP models using explanatory feedback on features, word-level attention, etc. Schramowski et al. (2020) highlight an intriguing interactive-learning-from-feedback setting where an expert in the loop gives feedback which can be used for similar counter-example- or gradient-based training. Stammer et al. (2021) extend this approach in neurosymbolic models to intervene on symbolically-conveyed semantics rather than purely visual features. In RL, however, applications of feature explanations have been more limited, although Guan et al. (2021) used human annotations of relevant visual features (together with binary feedback) to generate augmentations that varied the task-irrelevant features, and showed benefits over other feature-based explanation techniques or standard augmentations in video game playing.

**Language in RL:** Language is used broadly in RL, whether as instructions (e.g. Hermann et al., 2017; Kaplan et al., 2017), to target exploration (Goyal et al., 2019), or as an abstraction to structure hierarchical policies (Jiang et al., 2019). Luketina et al. (2019) review a wide variety of recent uses of language in RL, and argue for further research. However, they do not even mention explanations. Tulli et al. (2020) consider using natural language explanations of actions in RL. However, they only evaluate on a simple, symbolic MDP, and they do not observe benefits, perhaps because the explanations they use do not relate to the abstract structure of the task.

**Auxiliary tasks:** Predicting explanations is part of the general paradigm of shaping agent representations with auxiliary signals (e.g. Jaderberg et al., 2016). However, explanations are fundamentally different from unsupervised losses—unsupervised objectives are task-independent by definition, while explanations selectively emphasize the causally relevant features of a situation, and the relationship to general task principles (Lombrozo and Carey, 2006; Lombrozo, 2006). Some *supervised* auxiliary objectives are more similar to explanations; the boundaries of explanation are blurry. In the Alchemy environment (Wang et al., 2021), which involves learning latent causal structure, predicting task-relevant features improves performance. Similarly, Santoro et al. (2018) show that predicting a "meta-target"—an abstract label encoding some task structure—improves learning of a relational reasoning task. More broadly, supervising task inference can improve meta-learning (Rakelly et al., 2019; Humplik et al., 2019). Since these predictions directly relate to task structure, they are closer to explanations than unsupervised task-agnostic predictions. However, they do not necessarily actively link the details of the present situation to the principles of the task, as human explanations do.

## 5 DISCUSSION

We outlined the odd-one-out tasks, and showed that these tasks are challenging for RL agents to learn from reward alone. However, learning to generate language explanations significantly improved performance. Even though our agents lacked prior knowledge of language, learning to generate the language explanations helped them to discover the reasoning processes necessary for the task. Explanations helped agents to learn challenging and important abilities, such as relational and causal reasoning. This corresponds with human uses of explanations to identify the causally-relevant factors of a task, and thereby allow generalization (Lombrozo and Carey, 2006).

Because of this focus on task-specific structure, explanations outperform task-agnostic unsupervised auxiliary objectives. Indeed, we found that explanations helped agents to move beyond a fixation on easy shortcuts that do not fully solve the task, but that models nevertheless prefer (cf. Hermann and Lampinen, 2020; Geirhos et al., 2020). Furthermore, our results with confounded dimensions show that explanations can shape how an agent generalizes out-of-distribution (cf. Ross et al., 2017). Thus, explanations offer a promising route to training RL agents that learn and generalize better.

However, some care in crafting explanations is required to achieve their full benefits. Explanations must relate between the context, the agent's behavior, and the abstract task structure—explanations that ignore behavior are less useful, and those that ignore context are useless. Receiving explanations as input was not useful, likely because it is easier for the agent to ignore inputs than auxiliary targets. Explanations outperform unsupervised auxiliary reconstruction. Thus, simply training agents with more information (as with unsupervised objectives) is often not sufficient; explanations must provide relevant and specific learning targets to be most beneficial.

We also acknowledge that the boundaries of explanation are vague. For example, descriptions cannot name every property, so they tend to pragmatically focus on causally-relevant ones, and thus highlight similar features to explanations. This is why we refer to task-relevant property descriptions as explanations. Furthermore, we use "explanation" to refer to cues to relationships between specific situations, behaviors and abstract principles, which may overlap with other forms of auxiliary supervision. While we focused on language explanations, non-language predictions that highlight abstract task features could likely serve the same purpose. Explanations can also vary in abstraction (cf. Fyfe et al., 2014). The boundaries of explanation should be explored further in future work.

We also do not want to imply that explanations are *necessary* for learning. Most of the tasks we considered could potentially be learned with sufficient data alone, especially if combined with more complicated techniques, for example data augmentation (Raileanu et al., 2020; Guan et al., 2021), or auxiliary generative model learning (Gregor et al., 2019). Furthermore, many promising domains for deep learning—such as protein structure prediction (Jumper et al., 2021)—are precisely those areas that humans do not understand well, and so are challenging domains for humans to explain.

Indeed, some domains might be irreducibly complex; in these domains forcing a system to strictly follow simple explanations could be detrimental. Our approach does not force the agent to use explanations directly, and therefore might be less harmful in such cases than stronger constraints like requiring symbolic representations (Garcez and Lamb, 2020). We leave these issues for future work.

In other domains there may exist simple explanations that humans have not yet discovered. This observation motivates a future research direction: learning to explain over diverse task distributions, leveraging human explanations in domains we do understand. A curriculum focused on producing explanations could potentially yield substantial benefits. Humans generalize better after explaining, even without feedback (Chi et al., 1994; Rittle-Johnson, 2006), and this ability may be learned through education (cf. Nam and McClelland, 2021). An agent that similarly learns to produce explanations might similarly learn to generalize better *even in some domains for which we lack ground truth explanations*, and its explanations might help humans interpret its behaviour, and the domains.

**Conclusions:** We considered a challenging set of relational tasks, and showed that learning to generate explanations helps RL agents to learn these tasks across a variety of settings and paradigms. Explanations help agents move beyond biases favoring easy features, can determine how agents generalize out-of-distribution from ambiguous experiences, and can allow agents to meta-learn to perform experiments to identify causal structure. Because many of these abilities are thought to be challenging for current RL agents, we suggest that generating explanations as an auxiliary learning signal, rather than purely for post-hoc interpretation, may be a fruitful direction for further research.

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

## A ABLATION EXPERIMENTS & FURTHER ANALYSES

In this section, we perform a variety of control, ablation, and auxiliary experiments that identify which attributes of explanations are useful in different settings. We perform most of these experiments in the 2D RL setting because of the efficiency of running and training agents in this environment.

### A.1 AGENTS TRAINED WITHOUT EXPLANATIONS FIXATE ON THE EASIEST FEATURE DIMENSIONS

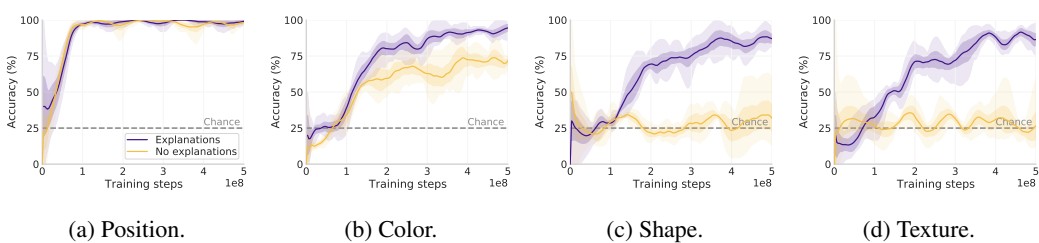

(a) Position.   (b) Color.   (c) Shape.   (d) Texture.

Figure 7: In the 2D setting, agents trained with explanations learn all dimensions, but agents trained without explanations learn to fully solve the tasks only if the relevant dimension is position (the easiest dimension), and only partly learn to solve the tasks with color (the next easiest dimension). (5 seeds per condition.)

In the basic 2D odd-one-out tasks, the agent achieves off-chance performance without explanations (while in more complicated settings such as the causal interventions, it cannot learn at all without explanations). In Fig. 7, we show that what the agent is doing is latching on to the feature dimension(s) that are most salient and *easiest* (Hermann and Lampinen, 2020), and only correctly solving episodes involving these features. Specifically, position is the most salient feature and is learned rapidly even without explanations, followed by color which is partially learned without explanations. However, shape and texture are much more difficult and are not learned well without explanations. These results concord with the features that Hermann and Lampinen (2020) found were easiest for CNNs and ResNets to learn, suggesting that explanations may help overcome the preference of agents (or other networks) to be "lazy" and prefer "shortcut features" (Geirhos et al., 2020).

### A.2 EXPLANATIONS ARE MOST USEFUL IF THEY ENGAGE WITH THE AGENT'S BEHAVIOR; SHUFFLED EXPLANATIONS ARE USELESS

We next investigate whether explanations need to be relevant to the agent's behavior, or even to the situation at all, in order to be useful. To do this, we provide the agent with explanations that either are situation-relevant, but behavior irrelevant, or are irrelevant to both behavior and situational context. To produce the situation-relevant but behavior-irrelevant explanations, we first construct an episode as before. We then enumerate all the property and reward explanations that it would be possible to receive in that episode, and present a randomly selected one to the agent on approximately 10% of steps, regardless of the agent's actions. These explanations do contain information about the objects in the scene, and can therefore potentially still benefit learning, but they do not directly react to the agent's actions.

We also considered context-irrelevant explanations that were randomly sampled from the set of all possible explanations (we chose either a property explanation or a post choice one with 50% probability, and then sampled a random set of attributes to fill out the template). This condition is essentially a control for the possibility that predicting structured information—even meaningless information unassociated with the task—could be acting as form of regularization.

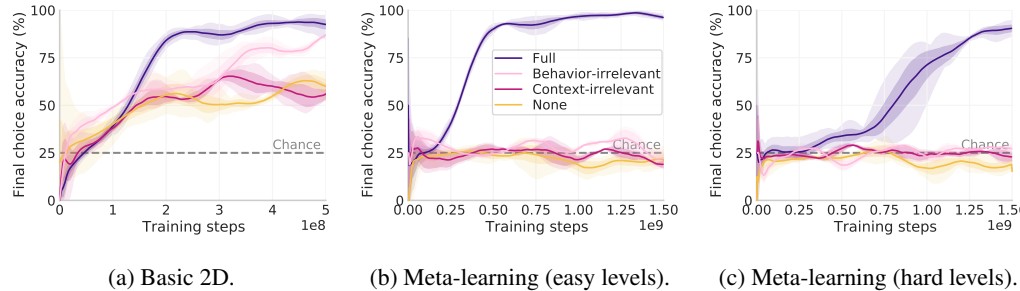

(a) Basic 2D.        (b) Meta-learning (easy levels).        (c) Meta-learning (hard levels).

Figure 8: Explanations must be behaviourally (as well as contextually) relevant to be useful in challenging settings; explanations that are contextually-irrelevant are useless in every experiment. (a) In the basic 2D odd-one-out tasks, behavior-irrelevant explanations eventually result in relatively comparable performance compared to full explanations, but produce much slower learning. Context irrelevant explanations are not substantially different than no explanations. (b-c) In both easy and hard learning-to-experiment levels, only an agent with full, behavior and context-relevant explanations is able to learn the tasks at all. Thus, more challenging task settings require more specific, behavior-releavnt explanations. (3 seeds for behavior-irrelevant/context-irrelevant conditions.)

Our results (Fig. 8) show that explanations that are relevant to both situation and behavior are most useful, situation-relevant but behavior-irrelevant explanations can be better than nothing in some cases, and totally irrelevant explanations are not beneficial at all. Specifically, for the basic tasks behavior-irrelevant explanations still result in some learning, but are much slower than full behavior-relevant explanations.

### A.3    PROVIDING EXPLANATIONS AS AGENT INPUTS IS NOT BENEFICIAL, AND INTERFERES WITH LEARNING FROM EXPLANATION TARGETS

In the main text, we focused on explanations as targets during training, rather than inputs. That approach is beneficial, because it does not require explanations at test time, while explanations as input generally does. In Fig. 9, we show that furthermore *providing explanations as input to the agent is not beneficial, and is actively detrimental if explanations are also used as targets*, presumably because in the latter case the agent can just "pass through" the explanations, without having to learn the task structure. However, it is possible that providing explanations as input on the timestep *after* the agent predicts them could be useful (as in language model prediction, where each word is input after the model predicts it); we leave evaluating this possibility to future work.

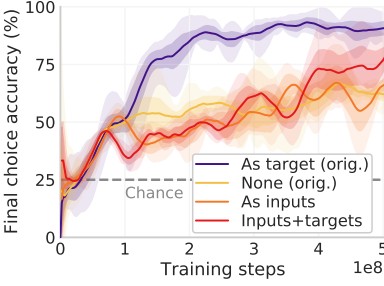

Figure 9: Providing explanations as agent inputs is not beneficial (performs better than no explanations), and is actively detrimental if explanations are also used as targets. (3 seeds per as-input condition, 5 seeds for main conditions.)

### A.4 DIFFERENT KINDS OF EXPLANATIONS HAVE COMPLEMENTARY, SOMETIMES SEPARABLE BENEFITS

We generally provided agents with both property explanations and reward explanations. Is one of these explanations more useful than the other? Are they redundant? To answer these questions, we considered providing the agent with each kind of explanation independently. We generally find that having both types of explanations is best, and the benefits of different types depend on the setting.

In the 2D setting (Fig. 6a) either type of explanations alone results in learning, but both types together result in substantially faster learning. In the 3D setting (Fig. 6b), we find that property explanations are uniquely beneficial; perhaps because predicting explanations on encountering the objects helps the agent overcome the challenges of generating good representations for its memory.

For the meta-interventions tasks involving experimentations, we find (Fig. 6c) that the reward explanations are uniquely beneficial, while properties explanations are not useful alone. The likely reason for this is clear when considering the episode structure—the relevance of a transformation to the final reward is much more directly conveyed by the reward explanations than the property ones. However, both types of explanations together are required for complete learning within the learning time we considered.

### A.5 THE BENEFITS OF EXPLANATIONS DEPEND ON TASK COMPLEXITY

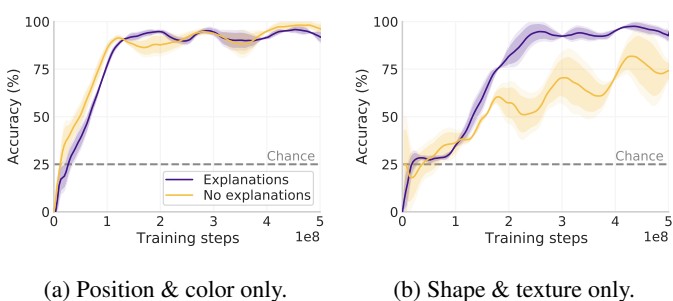

(a) Position & color only.                    (b) Shape & texture only.

Figure 10: The benefits of explanations depend on task difficulty. We train agents in the 2D environment on easier odd-one-out tasks where only two of the four dimensions are ever relevant. (a) When only the easy dimensions of position and color are ever relevant, the agents trained without explanations learn just as rapidly as the agents trained with explanations. (b) When the agents are trained on levels where only the harder dimensions of shape and texture are relevant, explanations still accelerate learning substantially. However, the agents trained without explanations achieve some learning in this condition, while they do not achieve any learning on these dimensions in the harder tasks used for the main experiments (see Fig. 7c-d). (2 seeds per condition.)

While we generally considered tasks with many feature dimensions that might be relevant, here we show that simpler tasks in which only two dimensions vary do not always require explanations for learning. However, task complexity depends on both the number of possibly relevant dimensions and the base difficulty of those dimensions. In Fig. 10 we show that explanations are not beneficial compared to no-explanations when only the easy features of position and color are relevant. Explanations are still beneficial when the features are more difficult (shape and texture). But even in this condition, the agent without explanations exhibits some learning, while it does not learn these dimensions at all in the main experiments, where the easier dimensions are also included (see Fig. 7c-d). Note also our results on deconfounding—in some cases explanations may help the agent to generalize in a desired way even if they are not necessary for learning the training task.

## A.6  LEARNING PROPERTIES THROUGH A CURRICULUM RATHER THAN AUXILIARY LOSSES

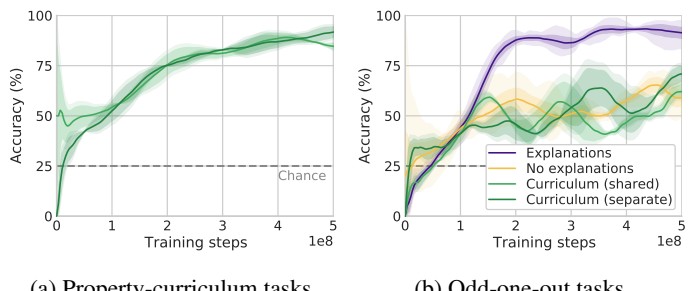

(a) Property-curriculum tasks.    (b) Odd-one-out tasks.

Figure 11: Agents trained with a curriculum of tasks that teach the properties (a) do not learn the odd-one-out tasks (b) any better than agents trained with no explanations. Results are similar whether the agent uses a shared policy (light green) for both the curriculum and odd-one-out tasks, or uses separate policies for each (dark green). (2 seeds per condition for curriculum conditions, 5 seeds for main conditions.)

Because predicting property explanations alone can be beneficial, we next consider whether the agent could benefit from learning properties through auxiliary tasks which teach those properties, rather than through explanations. Specifically, we provide the agent with a simpler property-learning task in 50% of episodes, where it receives a property like "red" as an input instruction, and has to choose the corresponding object (all objects are different along each feature dimension). These tasks provide a different way to force the agent to learn the properties of the objects. On the odd-one-out tasks, the agent receives the instruction "find the odd one out" to distinguish its goal.

Surprisingly, we find that learning these tasks does not substantially accelerate learning of the odd-one-out tasks compared to a no-explanations and no-property-tasks baseline (Fig. 11b). We initially thought this might be due to interference due to the shared policy being used for different tasks, so we reran the experiment with separate policy heads for the curriculum tasks and odd-one-out tasks, but this did not substantially change results. Auxiliary prediction of explanations may therefore be a more efficient way to encourage learning of task-relevant features, perhaps because it actively engages with the agent's behavior in the settings where those dimensions are particularly relevant.

## A.7  AUXILIARY UNSUPERVISED LOSSES ARE NEITHER NECESSARY NOR SUFFICIENT; THUS THE BENEFITS OF EXPLANATIONS ARE NOT SIMPLY DUE TO MORE SUPERVISION

In Fig. 12 we show that the auxiliary reconstruction losses are not necessary for learning the odd-one-out tasks. Furthermore, the main text results without explanations show that these losses are not sufficient for learning either. This shows that the benefits of explanations are not simply due to having more supervision for the agent, but rather are specific to supervision that highlights the abstract task structure.

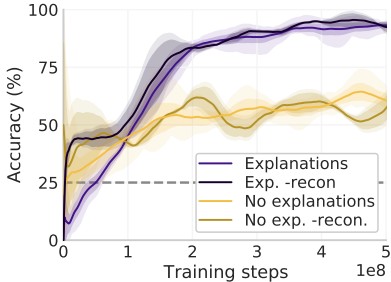

Figure 12: Ablating the auxiliary reconstruction losses does not alter the pattern of results in the 2D environment—thus reconstruction is not necessary for learning these tasks. Comparing to the results in Fig. 3b, which does include reconstruction losses, shows that reconstruction losses are also not sufficient without explanations. (2 seeds per condition.)

Table 1: Numerical results from main experiments/figures in each domain—mean $\pm$ standard deviation across seeds. Results are average performance (% correct) across evaluations during the last 1% of training.

| Experiment | Level | Fig. | Condition | Performance |
|---|---|---|---|---|
| Perceptual 2D | - | 3b | Explanations | $91.3 \pm 0.7$ |
| | | | No explanations | $61.9 \pm 2.2$ |
| Perceptual 3D | - | 3c | Explanations | $92.7 \pm 1.4$ |
| | | | No explanations | $29.5 \pm 0.7$ |
| Deconfounding | Chose color | 4b | No explanations | $55.4 \pm 2.6$ |
| | Chose shape | | No explanations | $24.2 \pm 7.6$ |
| | Chose texture | | No explanations | $15.4 \pm 6.7$ |
| | Chose color | 4c | Explain color | $95.5 \pm 0.9$ |
| | Chose shape | | Explain shape | $87.5 \pm 2.9$ |
| | Chose texture | | Explain texture | $86.2 \pm 0.9$ |
| Meta-learning | Easy | 5b | Explanations | $96.9 \pm 0.3$ |
| | | | No explanations | $24.0 \pm 0.6$ |
| | Hard | 5c | Explanations | $24.6 \pm 1.2$ |
| | | | No explanations | $61.9 \pm 2.2$ |

## B    Quantitative results

## C    Methods

### C.1    RL agents & training

Table 2: Hyperparameters used in main experiments. Where only one value is listed across both columns, it applies to both.

|  | **2D** | **3D** |
|---|---|---|
| All activation fns | ReLU | |
| State dimension | 512 | |
| Memory dimension | 512 | |
| Memory layers | 4 | |
| Memory num. heads | 8 | |
| TrXL extra length | 128 | |
| Visual encoder | CNN | ResNet |
| Vis. enc. channels | (16, 32, 32) | |
| Vis. enc. filt. size | (9, 3, 3) | (3, 3, 3) |
| Vis. enc. filt. stride | (9, 1, 1) | (2, 2, 2) |
| Vis. enc. num. blocks | NA | (2, 2, 2) |
| Policy & value nets | MLP with 1 hidden layer with 512 units. | |
| Reconstruction decoder | Architectural transpose of the encoder, with independent weights. | |
| Explanation decoder | 1-layer LSTM | |
| Explanation LSTM dimension | 256 | |
| Recon. loss weight | 1. | 0. |
| $V$-trace loss weight | 1. | |
| $V$-trace baseline weight | 0.5 | |
| Explanation loss weight | $3.3 \cdot 10^{-2}$ for main, 0.2 for meta-learning. | $3.3 \cdot 10^{-2}$ |
| Entropy weight | $1 \cdot 10^{-2}$ | $1 \cdot 10^{-3}$ |
| Batch size | 24 | |
| Training trajectory length | 50 | |
| Optimizer | Adam (Kingma and Ba, 2014) | |
| LR | $1 \cdot 10^{-4}$ | $5 \cdot 10^{-5}$ with explanations, $2 \cdot 10^{-5}$ without |

In Table 2 we list the architectural and hyperparameters used for the main experiments. In most cases the hyperparameters were taken from other sources without tuning for our setup. There are two main exceptions: (1) we chose the explanation weight loss to approximately balance the magnitude of this loss with the magnitudes of the RL losses early in training in each experiment, and (2) we swept the learning rate for the main experiments in 2D and 3D (but used the same settings for follow-up experiments, including meta-learning). In the 2D experiments we found that a similar learning rate was best for both agents trained with and without explanations, but in 3D we found that agents trained without explanations needed a slower learning rate to avoid their performance degrading from chance-level to below chance.

Since we ran the 3D experiments after 2D, we used similar hyperparameters, except that we found we needed to decrease the learning rate as noted above. However, some hyperparameters do differ across tasks due to specific task features. For example, the visual encoder for the 2D tasks is set to have a filter size of 9 because this is the resolution of each square in the grid, and the entropy cost for 2D tasks was chosen from prior work which used a similar grid world action space (Hill et al., 2019), while the 3D cost is lower because of the more complex action space for these tasks (see below). These decisions were shared across experimental conditions, so should not favor one condition over another.

**Explanation prediction loss:**    We trained the agents to predict the language explanation using a softmax cross-entropy loss over a word-level vocabulary of 1000 tokens (more than were necessary for the limited language we used). The explanation loss was summed across the sequence of tokens.

**Self-supervised image reconstruction loss:**    We trained the agents to reconstruct the image pixels (normalized to range [0, 1] on each color channel) with a sigmoid cross-entropy loss. The image reconstruction loss was averaged across all pixels and channels. However, we found in follow-up

experiments that this did not substantially change results in 2D (Appx. A.7), so we disabled this loss in 3D.

## C.2 RL ENVIRONMENT DETAILS

### C.2.1 2D

The tasks were instantiated in a $9 \times 9$ tile room with an extra 1 tile wall surrounding on all sides, for a total of $11 \times 11$ tiles. This was upsampled at a resolution of 9 pixels per tile to form a $99 \times 99$ image as input to the agent. The agent was placed in the center of the room, and had 9 possible actions, allowing it to move one square in any of the 8 possible directions, or to do nothing.

Four objects were placed in the room with the agent. They were chosen so that a single object was the odd one out, along a single dimension, and features appeared in two pairs along the other dimensions. The objects varied along the feature dimensions of:

- Color: one of 19 possible colors (e.g. green or lavender).
- Shape: one of 11 possible shapes (e.g. triangle or tee).
- Texture: one of 6 possible textures (e.g. horizontal stripes or checkers).
- Position: One of 4 position types (in corner, against horizontal wall, against vertical wall, or in a 3x3 square in the center).

The agent was given 128 steps to complete each episode, after which the episode would immediately terminate. The agent had to choose an object by walking onto the grid cell containing it. It would be immediately rewarded 1 if the object was the odd one out, and 0 otherwise. However, the episode would last for an additional few steps to give the agent time to learn from the reward explanation (if provided); this extra time was provided even if the agent was not trained to predict explanations, in order to precisely match the training experience across conditions. No additional reward would be received during this period, but the agent would be asked to output the reward explanation at every timestep. This period would last either 16 steps, until the agent touched any object, or until the full episode limit of 128 steps was reached, whichever came soonest.

If the agent was trained to predict property explanations, whenever it was adjacent to an object it would be asked to predict a string of the form:

```
This is a red horizontal-striped triangle in-the-corner
```

The properties always appeared in the order color texture shape position. These sentences were tokenized at a word level, with the hyphenated phrases treated as single words. Hence, a single token was attached to each possible feature value along each dimension.

If the agent was adjacent to multiple objects, which description it received was determined randomly. Once the agent made a choice, property descriptions were disabled.

If the agent was asked to predict reward explanations, for the period after receiving the reward (see above), it would be asked to predict a string in one of the following forms:

```
Correct because it is uniquely horizontal-striped

Incorrect because other objects are red horizontal-striped
triangles or in-the-corner
```

Thus, the reward explanations identify all features that contributed to a decision being incorrect.

### C.2.2 3D

The agent was placed in a room with a randomly placed door and windows (the agent could not interact with these). It had 10 possible actions: moving forward or backard, moving left or right, looking left or right or up or down, grabbing an object it was facing (if within a certain distance), and doing nothing.

Four objects were placed in the room, with attributes sampled as in the 2D environment from the dimensions:

- Color: one of 10 possible colors (e.g. blue or magenta).
- Size: one of 3 possible sizes (small, medium, or large).
- Texture: one of 6 possible textures/materials (e.g. metallic or wood-grain).
- Position: One of 3 position types (in corner, against wall, or in the center).

The episode lasted for 60 seconds, at 30 FPS; but the agent took an action only once every 4 frames (the action was then repeated until the next agent step), so the episode lasted for at most 450 agent steps. The agent had to use its "grab" action on an object to make a choice; colliding with the objects would simply cause them to move. 25 steps were allocated for reward explanations (if any).

If the agent was asked to predict property explanations, they were given when the agent was facing an object and close enough to grab it.

The property and reward explanations for the 3D environment were analogous to the ones for the 2D environment, except that the reward explanaitons did not have the "or" before the final attribute.

