# OpenReview forum: "Tell me why!—Explanations support learning relational and causal structure"
_ICLR.cc/2022/Conference — ICLR 2022 Submitted_

### Official Review · Reviewer_x1cM · 2021-10-29

**Correctness:** 3
**Technical Novelty And Significance:** 1
**Empirical Novelty And Significance:** 3
**Recommendation:** 6
**Confidence:** 4

**Main Review:**

I enjoyed reading this paper, and thank the authors for their clear exposition.

The intro and background were clear, and well-situated in the computer and cognitive science, and the psychology literature. The experimental setup was also one I had not seen before (the authors say it is one of their contributions on page 2; so thank you for this interesting setup).

Despite the authors' good grasp of the state of the art, and in fact, because it, I would ask that the authors refrain from over-claiming in their experimental results. As this paper will likely be read by many researchers, I would emphasize the potential for impact that the paper has to demonstrate proper drawing of scientific conclusions. Some examples here:
* bot of page 3: "never perform above chance"; they do, in the middle and towards the end of training --> "barely/hardly/seldomly perform above chance"
* top of page 5: "allows strong ood generalization"; what is strong? have you quantified this? even qualitatively/relatively speaking, one needs some metric to compare diff settings.
* mid of page 8: "causally-relevant and generalizable explanations": what is meant by this? causality has clear definitions, which are absent here. on the other hand, explanation relevance is difficult to ascertain (as the authors have mentioned), thus "causally relevant explanations" is doubly vague here. Furthermore, causal relations, at least in the SCM framework and through the independent mechanisms assumptions, is considered "generalizable" by definition.
* mid of page 9: "Generally, having more types of explanations is better"; this is false, as you had shown that random/irrelevant explanations hurt performance. In your investigated setting, and for the two types of explanations, more = better most of the time.
* title of App A.3: the wording of "actively detrimental" is misleading, or confusing at best, and this depends on what you compare with. If explanations are inputs, or input+targets, then there is no change compared to having explanations at all. Explanations only help (in the context of your experiment) if they are only the targets (and relative to this setting, adding them additionally as inputs is detrimental, yes, but one naturally expects the baseline to be the case presented in the abstract/intro, where explanations are absent everywhere).

Clearly, the paper investigates many empirical setups, which in the end show that explanation targets are useful in various settings. I wonder:
* the training objective (not seen in the main body) comprises of various parts; the explanation part is essentially a regularizer, no?
* are there settings in which agents don't require 10^8 (in 2D) or 10^9 (in 3D and in causal 2D) training steps to start performing?
* is human language the best regularizer? are there other parsimonious languages that would better suit this task (and perhaps reduce the number of training steps in the process)?
* curious why you chose to have an explicit agent to explore and environment to do the odd-one-out task; this adds more complexity to the system and makes the subsequent analysis of the impact of explanation targets more difficult.
* how can one compare the effect of the presented auxiliary loss of explanability with other auxiliary losses?

**Summary Of The Paper:**

This paper investigates the use of explanations during the training of an agent for downstream tasks. The investigated tasks are variants of the odd-one-out setting, and the experiments demonstrate improved performance on this task (relative to training without explanation targets) for static (observational) and interactive (interventional) settings when the agent is trained with description-based and/or reward-based explanations. Additional experiments are performed to empirically show the ability of such an agent to "deconfound" the shape, color, texture, and sometimes size of the objects, to study the effect of random/irrelevant explanations, and to perform various other ablation studies.

**Summary Of The Review:**

The exposition is clear and the background and relevant literature are well-represented. The paper only lightly presents a theoretical analysis of the work, and the value of the work seems to be primarily based in the comprehensive set of experiments presented, which serve to show that training an agent with an auxiliary loss to generate explanations (alongside a scene reconstruction and policy optimization) would allow the agent to perform better on various odd-one-out settings, compared to had it not been trained to give explanations. Notwithstanding over-claimed statements (to-be-addressed) and some open discussion questions highlighted above, this paper seems like one that the ICLR would enjoy reading and referring to.

---

> ### Author Response · Authors · 2021-11-17
> **We thank the reviewer for their thorough review, and have corrected the issues and revised the framing**
>
> We thank the reviewer for their careful review, we are glad to hear they enjoyed the paper! We have revised the article to try to state our results and conclusions more precisely. We also have more generally reframed the article, especially the introduction and conclusions to reflect these comments and those of other reviewers. Changes include:
>
> * Amending the claims noted, as well as a few others throughout. For example, we have changed the “causally-relevant and generalizable explanations” sentence to “This corresponds with human uses of explanations to identify the causally-relevant factors of a task, and thereby allow generalization (Lombrozo and Carey, 2006).”
> * Focusing introduction and discussion more on RL, and discussing why RL is particularly interesting (e.g. in the second paragraph of the introduction): First, because RL agents are able to perform interventions (i.e. to *do* in Pearl’s framework) and interact over time, and so provide a unique testing ground for learning of causal relationships. Second, because the sparse learning signals in RL, together with the challenges and complexity noted in this review, also makes these settings more likely to benefit from explanations.
> * Highlighting the auxiliary unsupervised reconstruction objective and other control conditions we compare to, which show the benefits of more specific explanations; but also acknowledging that other auxiliary objectives that focus more on structure, such as auxiliary learning of generative environment models [Gregor et al., 2019] could potentially provide some of the benefits of explanations. This also relates to the distinction between explanations and a regularizer—a regularizer is generally task-agnostic.
> * Clarifying in the caption of the first figure that “Training steps” refers to agent/environment steps, while the number of parameter updates is roughly 10^4 times smaller. It’s worth noting that the agent is learning to interact with objects (in order to have a 1/4 chance at reward) quite early; but discovering the full reasoning process takes longer precisely because the credit-assignment problem is difficult. We do have some simplified versions of the task (not included in the paper at present) where the inputs are more discrete (a restricted set of possible discrete inputs rather than pixels) and the agent sees a single object at each time step without having to explore. In these simpler settings, the agent learns substantially within 10k updates or so. If it would be helpful, we would be happy to add these results to an appendix.
>
> We thank the reviewer for taking the time to review our paper so carefully, and we hope that they will find that this version addresses their questions and concerns.

---

### Official Review · Reviewer_DFpa · 2021-11-02

**Correctness:** 3
**Technical Novelty And Significance:** 2
**Empirical Novelty And Significance:** 2
**Recommendation:** 6
**Confidence:** 2

**Main Review:**

The paper is mostly a position paper supported by dedicated experiments in specific setups intended to support the various claims. I was not familiar with sort-one-out identifications, but I have found the arguments of the authors convincing. Similarly the discussion about the related works seems to be quite detailed. Having this in mind, my educated guess about the paper is that it might be interesting to have it presented at ICLR. A weakness I noticed is the very dry description of the experimental setup.On the other side, the rest of the presentation is very clear and easy to follow. I should also notice that the claims are mostly supported by empirical analysis in very specific environments, deciding whether or not this is enough to claim a general validity could be controversial.


**Summary Of The Paper:**

This is a paper about sort-one-out identification. The goal is to emphasise that standard reinforcement learning is not optimal for this task, while adding (language) explanations to the learning phase might be highly beneficial. Dedicated experiments are developed for 2- and 3-D simulation environments. Each experiment is used to advocate a specific point such as the effect of the explanations on accuracy, robustness wrt correlated features, and other more detailed analyses.

**Summary Of The Review:**

A well-written paper advocating the advantages of adding explanations to RL setups wrt to sort-one-out identifications. As a non expert of this field I have found the presentation clear and convincing. The arguments to support the claims are mostly empirical, but restricted to a specific simulation architecture.

---

> ### Author Response · Authors · 2021-11-17
> **We are glad the reviewer found the paper clear and convincing, and have tried to address their concerns**
>
> We thank the reviewer for their thoughtful and generally positive comments. We are glad to hear that they found the paper clear and convincing. We have added some more detail and edited the experimental setup presentation, which we hope will make it slightly more engaging. We have also reframed the paper to clarify our contributions and situate it better within the existing literature.
>
> Although our present results are limited to “odd-one-out”-style environments, we note that this type of task is easily generalizable across a variety of domains, such as 2D, 3D, or even text-only or other modalities. Odd-one-out tasks are particularly well-suited for testing relational abstraction faculties, being directly inspired by the cognitive science literature—where they have been thoroughly validated—while also being distinct from other deep RL benchmarks. Furthermore, we explored several settings beyond basic RL, such as the deconfounding setting (where reward alone cannot discriminate between the features), and the meta-learning paradigm, which demonstrate other dimensions of generality. We now better emphasize these contributions in the revised introduction, as well as mentioning them in the odd-one-out tasks section.
>
> We hope that the reviewer will find that these changes have improved and clarified the paper, and addressed the weakness highlighted in their review, as well as comments from the other reviewers.

---

### Official Review · Reviewer_AvzX · 2021-11-02

**Correctness:** 1
**Technical Novelty And Significance:** 1
**Empirical Novelty And Significance:** 1
**Recommendation:** 3
**Confidence:** 4

**Main Review:**

Despite the idea of including textual explanations in the learning pipeline to improve performance is interesting and trending these days in the DL research community, the overall contribution of this paper is very modest.
The proposed odd-one-out tasks do not cast a significant new challenge with respect to existing tasks. Authors justify their introduction because they require relational reasoning and abstraction, described as "identifying uniqueness requires reasoning over all objects, and all dimensions along which objects may be related." Yet, existing tasks already cover such requirements, e.g. CLEVR [1].
The paper's central claim -namely that learning to generate explanations is useful to solve relational tasks - is not adequately supported. In this regard, there are two major concerns:
- What are explanations? To give substance to a claim involving *explanations*, these should be rigorously defined. This definition is missing. Moreover, the proposed *property explanation (PE)* clashes with the intuitive idea of explanation, PEs resemble more a description. Can we consider any additional information about objects as an explanation? Should this information be structured in a particular way to gain the status of explanation? If yes, how? If not, what is the point of introducing the concept of explanation?
- Experiments show a comparison between an agent trained using only input images and an agent trained also using additional relevant information about the properties of each object. This setup could only say something about the performance of agents trained using more or less information. Instead, we expect to see the comparison between agents trained using the same amount of information, but in one case shaped as an explanation and in the other case not. It is not clear what the authors mean by "prior knowledge of language."

The paper is not properly structured. The methods section is far too short. Many aspects should be expanded, such as the description of the RL environment and the training datasets for the NN.
The experiment section is poor despite its length, it should be enriched with a comparison against existing approaches, even if slightly out-of-task for odd-one-out. The related work section suffers the same problem. It is far too long and, in the end, not very informative for the proposed approach.
Section 4 provides interesting references to literature in psychology and neuroscience. Yet, it should be shortened by removing less relevant parts, such as the "Self-explanation" paragraph.
Section 5 does not do a good job in placing this work in AI literature. References are too generic (e.g. the NLP paragraph) and should tackle more competitive approaches.

[1] Justin Johnson, Bharath Hariharan, Laurens van der Maaten, Li Fei-Fei, C. Lawrence Zitnick, and Ross Girshick. CLEVR: A Diagnostic Dataset for Compositional Language and Elementary Visual Reasoning. In Proc. CVPR, pp. 1988–1997, 2017.

**Summary Of The Paper:**

The authors introduce 2D and 3D Reinforcement Learning tasks that ask the agent to find the odd object in a collection of objects. The experiments show that an agent trained using only input images performs worse than an agent trained also using clean abstract information about the properties of each object.

**Summary Of The Review:**

Despite the idea of including textual explanations in the learning pipeline to improve performance is interesting and trending these days in the DL research community, the overall contribution of this paper is very modest. The paper's central claim is ambiguous due to a missing operative definition of explanation and not adequately supported by experiments.

---

> ### Author Response · Authors · 2021-11-17
> **We have tried to clarify these issues in the revised version of the paper (1/2)**
>
> We thank the reviewer for their feedback and comments. Their rating seems to be mainly based on the following criticisms: 1) lack of an operative definition of "explanations", and 2) not controlling for the amount of information provided to the agent. As we'll detail below, we think these concerns are easily addressable or are based on misunderstandings (which we have tried to clarify in our revision).
>
> What is an explanation?
> * We respectfully disagree that a definition of explanations is missing—in the introduction we state that: “We define a language explanation to be a string that indicates relationships between a situation, the agent’s  behavior, and abstract task structure.“ We have reorganized and elaborated to emphasize and clarify this definition. While a more rigorous formal definition would be ideal, it is not clear whether it would be feasible––the definition of explanation has been debated in philosophy and cognitive science for hundreds of years, and we reference a variety of works from that literature to emphasize the various perspectives presented and motivate the definition we use. We also discuss these issues explicitly in the discussion section; in particular we state why we consider “descriptions” to sometimes satisfy the properties of explanations. Furthermore, our experiments demonstrate a variety of empirical observations on the boundaries of explanation (see next point), which we hope will be useful for articulating a more rigorous definition of explanations in future work.
>
> Amount of information provided to agents.
> * The main point of our paper is that the *type* of information provided by explanations could be useful for learning and generalizing. Indeed, the information provided in the explanation is already contained in the visual observations, so our experiments are really about the type of information conveyed and the method for conveying it (see next point). Furthermore, there are many RL papers published in ICLR and other conferences that consider ways of using auxiliary information to help learning, such as demonstrations or expert trajectories, human trajectory preferences or reward sketches, etc. From this perspective, it seems that highlighting a useful auxiliary source of information is a valid contribution. Furthermore, as the cognitive literature we review shows, explanations are potentially a *particularly* useful form of additional supervision.
> * Many of our experiments are not simply about the amount of information provided, but the type of information or the way in which that information is provided. For example, agents that receive the same explanations as input rather than targets show no benefits (A.3), yet they receive the same amount of information as our main explanation-as-targets condition. We further show that explanations that refer to objects in the scene but ignore the agent’s actions (and so convey nearly the same amount of information) are much less effective (A.2 Fig. 8). We discuss several times in the text that agents trained with unsupervised objectives that in some sense provide more information (in terms of number of steps supervised, or minimum description length of each supervisory signal) do not receive nearly as great of benefits (A.7 Fig. 12, to which we have added the main text results in the revision to make the contrast more salient). We also show that teaching the agent the properties through curriculum tasks is not as effective as explanations, even with a separate policy to avoid interference (Section A.6). Thus, our contributions illustrate at least some aspects of what type of information is useful, how it should be structured, and how to convey it to the agent. We have tried to emphasize these points in the revised discussion.
>
> (continued in next comment.)

---

> > ### Author Response · Authors · 2021-11-17
> > **We have tried to clarify these issues in the revised version of the paper (2/2)**
> >
> > (continued)
> >
> > Comparing to other approaches:
> > * We do make a variety of comparisons, some of which are outlined above. But our contributions are more of an argument for a different way of approaching the problem of achieving desired behavior. It’s not clear what existing approaches would change the interpretation of our results—as we acknowledge in the discussion, we are not saying that explanations are *necessary* for solving tasks, merely that they can be useful (and may be relatively easy to collect from humans, compared to e.g. collecting demonstrations).
> >
> > Why not use an existing task like CLEVR?
> > * CLEVR is a supervised learning task, and we focus on RL. The challenges are fundamentally different; for example, CLEVR does not require the model to perform intervention experiments to discover information, as our experiments in Section 3.4 do. Furthermore, CLEVR generally explicitly states the relations required in the question (e.g. “are there an equal number of red balls and metal cubes?”) whereas the odd-one-out tasks require the agent to consider many possible relationships in each episode, which may make learning more challenging, especially in the RL setting.
> >
> > Organization:
> > * We have moved some of the environment details to a new subsection of the tasks section (Section 1.1), and expanded the methods of agent training slightly.
> >
> > We have reorganized and reframed the paper to address these comments and those of other reviewers, and we tried to emphasize the main points listed above. We hope the reviewer will find the revised version more satisfactory, and we look forward to any additional feedback.

---

> > > ### Comment · Reviewer_AvzX · 2021-11-29
> > > **The paper has improved, the misuse of the word explanation remains**
> > >
> > > Having read the other reviews, I have realized the novelty of the RL environment. I also believe that the last revision of the manuscript marks an improvement over the original version of the paper. The new paragraph about the environment and the new introduction make the setting and placing of the work more transparent. I also appreciated the slight adjustments on the methods section, which is now decent, and the renaming of sections A.3 and A.7, which are more explicative.
> > >
> > > Yet, my main perplexity remains. What do we see in these experiments? That agents provided with additional high-level abstract information about the scene perform better than agents deprived of it.
> > > But what makes this additional information an explanation?
> > > The authors define in the introduction "*a language explanation to be a string that indicates relationships between a situation, the agent's behavior, and abstract task structure.*" The very first sentence of the paper is "*Explanations—language that provides explicit information about the abstract, causal structure of the world—are central to human learning*". Yet, these definitions are not operative: both emphasize the language nature of explanations, but there is no trace in the experiments where this nature is critical. Would we have the same improvement if the agent tries to predict a set of features organized as a vector rather than a language sentence?
> > >
> > > I updated my vote because the experiments show interesting phenomena, and the last version of the paper effectively reports them. Yet, I believe that when the main selling point of a paper is "explaining to an agent improves its performance", a precise and operative definition of what an explanation is and what explaining means is needed. I do not call for a universal philosophical definition, which is clearly out of the scope, but at least for a tentative association of the explanation concept to a mathematical object, which solves the task presented in the paper and could turn useful for other works in the ML area. I think this is what one expects when starting to read this work after having read the catchy title "Tell me why!—Explanations support learning relational and causal structure".

---

> > > > ### Author Response · Authors · 2021-11-29
> > > > **We thank the reviewer for their response, and have some further follow ups.**
> > > >
> > > > We thank the reviewer for taking the time to engage with our response, and we are glad that the reviewer finds the paper improved. We wanted to follow up on the reviewer's remaining concerns:
> > > >
> > > > First, we do explicitly address the issue of language vs. other features in the discussion: "While we focused on language explanations, non-language predictions that highlight abstract task features could likely serve the same purpose." The motivation for nevertheless focusing on language is two-fold:
> > > >   1) To connect with the cognitive science literature, because human explanations rely on language.
> > > >   2) Because of the previous point, language provides a natural and expressive medium for humans to interact with RL agents or ML systems more generally. As we note in the introduction: "Language explanations may be simpler for humans to produce than other forms of supervision (e.g. Cabi et al., 2019; Guan et al., 2021), and could selectively identify the key generalizable
> > > > features of a situation." Furthermore, as we note aspirationally in the future directions "[an agent's] explanations might help humans interpret its behaviour, and the [problem] domains it explains."
> > > >
> > > > We hope that this clarifies our motivation, and we would be happy to emphasize earlier in the text that language per se is likely not necessary at a computational level, merely convenient for humans.
> > > >
> > > > Second, we still believe that our experiments provide informative constraints on the definition of explanations. For example, the experiments we have already highlighted in our prior response clearly show that explanations must be behaviourally relevant to be useful, at least in the cases we considered, which is why our definition highlights the agents behavior. In addition, we show that, for example, explanations as input do not offer significant benefits, thus providing some constraints on how explanations should be used. While a formal definition could potentially be useful, we do not believe it is the only way for our results to be generalizable or beneficial to the community. Our goal is to inspire other researchers to explore similar approaches. If they find similar effects, that would offer a broader foundation on which to build a formal definition. Often progress in machine learning has preceded the formal explanation of it; indeed, what is there for theory to explain without empirical work? We hope that our empirical results, together with future research, could provide the inspiration for future formal developments.
> > > >
> > > > However, if there are aspects of the framing that we could alter to avoid setting up an expectation of a formal definition, we would be happy to do so. We attempted to write the abstract of the paper in a way that clearly states that our results are based on empirical experiments rather than formal definitions, but if the reviewer has specific suggestions that would ameliorate this expectation, we would be happy to consider them.

---

### Official Review · Reviewer_8zdV · 2021-11-04

**Correctness:** 4
**Technical Novelty And Significance:** 2
**Empirical Novelty And Significance:** 2
**Recommendation:** 6
**Confidence:** 3

**Main Review:**

I really enjoyed reading this paper.  It is very well written, most concepts are
presented clearly, and it explores an important research direction -- namely
how explanations provide invaluable supervision during learning.

The paper has two main downsides:

- It ignores most literature on learning from explanations.  The authors cite
the works of Camburu et al and Mu et al, but that's about it.  They go as far
as calling these "exceptions".  Quoting: "most recent work in AI neglects explanations as a learning signal."

This is however false.  In the last few years, the literature on this topic has
grown considerably and I could consider it unfair to not mention at least the
following works:

- Stammer, Wolfgang, Patrick Schramowski, and Kristian Kersting. "Right for the Right Concept: Revising Neuro-Symbolic Concepts by Interacting with their Explanations." In Proceedings of the IEEE/CVF Conference on Computer Vision and Pattern Recognition, pp. 3619-3629. 2021.

which specifically studies learning from explanations in the context of relational
prediction tasks.  The problem of "where do the explanations come from", namely
from human supervisors, has been studied in detail here:

- Schramowski, Patrick, Wolfgang Stammer, Stefano Teso, Anna Brugger, Franziska Herbert, Xiaoting Shao, Hans-Georg Luigs, Anne-Katrin Mahlein, and Kristian Kersting. "Making deep neural networks right for the right scientific reasons by interacting with their explanations." Nature Machine Intelligence 2, no. 8 (2020): 476-486.

The authors write "explanations could be produced by humans, e.g. as annotations of past trajectories" (p 3) and this is *exactly* the setting that the above paper studies.  This
work is based on an earlier paper:

- Ross, Andrew Slavin, Michael C. Hughes, and Finale Doshi-Velez. "Right for the right reasons: training differentiable models by constraining their explanations." In Proceedings of the 26th International Joint Conference on Artificial Intelligence, pp. 2662-2670. 2017.

where the authors develop a way to integrate supervision on saliency maps.  This
is very relevant although different from the linguistic explanations used in this
work.  The literature on debugging NLP models using explanations has been surveyed
in:

- Lertvittayakumjorn, Piyawat, and Francesca Toni. "Explanation-Based Human Debugging of NLP Models: A Survey." arXiv preprint arXiv:2104.15135 (2021).

Works on this topic already study whether and why explanations are useful for
debugging and deconfouding learned models.  Finally, concerning reinforcement
learning, a key paper is:

- Guan, Lin, Mudit Verma, Sihang Guo, Ruohan Zhang, and Subbarao Kambhampati. "Explanation augmented feedback in human-in-the-loop reinforcement learning." arXiv preprint arXiv:2006.14804 (2020).

Notice that I am *not* claiming that the current work is completely subsumed
by existing studies;  but I am saying that the authors neglec a large set of
works and that this is not acceptable.

The authors should at the bare minimum distinguish their original contributions
from those that were already made in this literature.

A more complete list of works on using explanations as supervision can be found
here:

- https://github.com/stefanoteso/awesome-explanatory-supervision

Besides this complaint, I liked the contents of the paper.  The main issue is,
to my eyes, that little is done to position the proposed work against existing
research.


** POST-REBUTTAL UPDATE **: The authors substantially improved their coverage of the related work.  I have increased the score accordingly.

**Summary Of The Paper:**

The authors introduce a meta-architecture for learning from explanations.  This
architecture is showcased by solving different instantiations of the same
relational task ("one-odd-out", identifying the one object that stands out
in a set of objects).  Empirical results showcase the usefulness of learning
from explanations and the ability of the proposed approach to do so.

**Summary Of The Review:**

Explores an important research direction but neglects existing literature.

---

> ### Author Response · Authors · 2021-11-17
> **We thank the reviewer for these resources, and have updated the paper accordingly**
>
> We thank the reviewer for their thoughtful review, and are glad to hear they enjoyed reading the paper! We appreciate their drawing our attention to this relevant literature, as well as sharing the literature list from Stefano Teso, which is an excellent resource.
>
> * We have incorporated the papers they mentioned as well as some other relevant work from the github list referenced (e.g. “Learning from Explanations and Demonstrations: A Pilot Study”, Tulli et al. 2020). These can be found in the related work, mainly in a new paragraph on explanations and revised paragraph on RL, as well as some references in the revised introduction and discussion.
> * To incorporate this literature, and address some of the questions raised by several of the reviewers, we have revised the introduction and discussion to attempt to better situate this paper against the prior work and to clarify why we think the topic is relevant, and why RL is an interesting setting.
> * For example, we have replaced the “neglects” framing in the introduction with the following: “Thus, there has been increasing interest in using explanations as learning signal (e.g. Ross et al., 2017; Mu et al., 2020; Camburu et al., 2018; Schramowski et al., 2020; see related work) [...] However, most research on learning from explanations has been within supervised learning. There have been a few explorations within reinforcement learning (RL; Guan et al.,2021; Tulli et al., 2020), but these have not focused on relational or causal reasoning.” We then go on to address some particular reasons why RL is a particularly interesting domain for investigating explanations and causality—namely due to the sparse learning signals, and the ability to learn causal structure.
>
> We hope that these changes have improved the paper and look forward to any further feedback, and thank the reviewer again for taking the time to craft such a helpful review.

---

### Author Response · Authors · 2021-11-17
**General response to the reviewers**

We thank the reviewers for their helpful and thoughtful feedback. Many reviewers agreed on the clarity, enjoyability, and interest of the paper for ICLR, although they had the following main concerns:
1. Clarifying the framing of the paper and highlighting why we chose to work on the tasks we did (for example, why work in RL?).
2. Better contextualization within the existing literature on explanations.
3. Controlling for the amount of information provided to the agent.
4. More clarity on the definition of explanations.
5. Better presentation of methods.

In response, we’ve made the following edits:
1. Substantially edited the introduction of the paper, and the related work section, to situate better within the prior work.
2. Correspondingly, highlighted the interest and merit of our particular perspective and investigations, including articulating explicitly why RL is a particularly relevant domain for investigating these issues.
3. Reorganized to emphasize the definition of explanations that we provided in the introduction, and adding slightly more language relating experiment results back to this definition.
4. Highlighting further within the text (and by editing the figure in Appendix A.7) the ways in which explanations provide a *specific* type of information in a *specific* way, and the controls we run which provide the same or more supervision, but in less useful ways.
5. Moved main environment details to a new subsection (Section 1.1) of the tasks section, and made various clarifications to the methods.

We hope reviewers agree that these changes have substantially improved the paper overall.

---

### Decision · Program_Chairs · 2022-01-20

**Decision:**

Reject

**Comment:**

This paper studies the use of natural language explanations during the training of an agent for odd-one-out tasks. Experiment results show that using quality explanation as abstract information about object properties helps with the agent performance, as compared with the vanilla method.

Strengths:
- Experiment results are conducted thoroughly to support the major claims made by the paper
- The problem is well motivated and has an important implication

Weakness:
- There has been extensive discussion about whether the paper lacks a more formal and rigorous definition of "explanation" as considered in the scope of this paper.
- Concerns are raised regarding the gaps between the broad claims in the paper and the restricted experiment settings